# Developing the Emotional Intelligence of Millennial Students: A Teaching Strategy

Oana-Andreea Ghita-Pirnuta [1] and Laura Cismaru [2,*]

1    Faculty of Letters, Transilvania University of Brasov, 25 Eroilor Bd., 500030 Brasov, Romania
2    Faculty of Food and Tourism, Transilvania University of Brasov, 148 Castelului St., 500014 Brasov, Romania
*    Correspondence: laura.cismaru@unitbv.ro; Tel.: +40-720289007

**Abstract:** The contemporary world is facing a real crisis due to the depletion of basic resources, consumerism and the unprecedented proliferation of physical and mental diseases that are caused by unhealthy nutrition and eating habits. The training of future specialists in the food industry with a high level of emotional intelligence, responsible both for themselves and for others, can be a useful strategy for getting out of the crisis. This was the general objective of the present research, whose statistical hypotheses aimed at increasing the level of empathy of students enrolled in two academic programs of study in the food industry by introducing four consumer psychology modules targeting emotional awareness and moral identity. The applied experimental treatment consisted of the students going through the activities of the four modules, with each module having a triarchic structure: teaching–self-testing–reflective writing. The design used was quasi-experimental, with a single group taking a pretest–posttest measurement. The results obtained support previous research efforts that had the same objective—to contribute to increasing millennial students' levels of empathy and awareness of their own emotions. Additionally, using the pedagogical techniques of self-testing and reflective writing proved useful in achieving the set objective.

**Keywords:** emotional intelligence; millennial students; transformational education; empathy; emotional self-awareness; moral identity; reflective writing

## 1. Introduction

The modern world is facing multifarious problems caused by wrong/unhealthy human nutrition, related to both the low quality of consumed food and consumption patterns. The increasingly aggressive marketing used in the food industry combined with consumers' lack of education concerning a responsible diet have, unfortunately, majorly contributed to the chronicle of problems related to unhealthy nutrition. International statistics show a dramatic rise in cancer, cardiovascular disease and depression, especially in people under 50, almost worldwide. This alarming situation has been addressed by experts and researchers in medicine, and one of the most common findings is related to the impact of a poor diet on cancer, depression and cardiovascular disease. For instance, more and more studies have linked highly and ultra-processed food to cancer, heart failure, obesity, depression and death. Young people are especially vulnerable because they have a high level of brand sensitivity and a low budget for food; therefore, they prefer fast food, which is not the most responsible and healthy diet choice, but very aggressively promoted. If consumers are less and less preoccupied with healthy nutrition, it should be up to food producers to act more responsibly and help change the wrong and unhealthy diet patterns of populations. In this context, concerns about training a responsible labor force have been acutely raised in the food industry, with specialists sincerely concerned about consumers' health, the high quality of the offered products, the environment, as well as shaping some healthy long-term consumption patterns. In the present generational context, in which the millennial generation is the most strongly represented in the labor

market, there is a question of what skills they need to obtain through their education, so that the above-presented aim can be fulfilled. The profile of millennials contains contradictory elements, such as possessing social responsibility, a high stress tolerance, an expertise in teamwork, but also lacking soft skills and being addicted to trends and consumerism, etc. Such a profile is very challenging from the perspective of those responsible for training and educating millennials. In this context, the present experimental research aims at testing the effects of introducing several consumer psychology modules on the level of emotional intelligence of the students enrolled in two programs of study in food engineering, the long-term aim of this approach being to train a more responsible workforce in the food industry. The innovative aspect of this approach consists of the inclusion of reflective writing tasks based on the students' psychological self-testing within the framework of the consumer psychology modules.

In the academic year 2020–2021, aware of the increasing importance of aspects related to consumer psychology in the food industry, the Romanian higher education authorities included a course entitled 'Psychology of Human Nutrition' in the list of possible subjects in higher education food industry programs adopted at the national level. Within the framework of the Faculty of Food and Tourism, which is part of Transilvania University of Brasov, it has been decided to run the course for fourth-year undergraduate students from food engineering majors. The 'Psychology of Human Nutrition' course has been a real success, with the students expressing their gratitude at the completion of their undergraduate studies, and 70% of the participating students ranking it first, second, or third out of their courses. Unfortunately, in the academic year 2021–2022, the subject was excluded from the national list, and the course was no longer taught in Romania. Based on the students' level of appreciation, in the Faculty of Food and Tourism, it has been decided to introduce consumer psychology modules within the framework of the Marketing course, taught by a staff member with a double specialization both in Marketing and Psychology. In this context, the decision has been made to carry out a study with an experimental design with its main aim to identify the positive effects of completing the psychology modules focused on the development of emotional intelligence in the case of the students enrolled in the Food Engineering programs. The obtained results support the conclusions of previous studies in the sense that they show the contribution of psychology modules and reflective strategies to the improvement of some basic components of emotional intelligence—in the case of the participating students, namely, empathy and awareness of their own emotions. Consequently, firstly, the present research aims at offering an example of good practices in the training of future specialists in the food industry. Furthermore, the research aims to highlight the importance of including applied psychology components related to emotional intelligence in the training of millennial students, as the main actors in the future workforce of this field.

*1.1. Emotional Intelligence and Millennial Students*

The concept of emotional intelligence was used for the first time in the early 1990s, being defined as a form of social intelligence which involves the ability to monitor and understand emotions, including one's own emotions and other people's emotions, in order to adapt thinking and behavior [1]. In the last two decades, the concept of emotional intelligence has gradually turned into a very popular one and has benefited from increasing attention from specialists, who have formulated different definitions and explanatory theoretical models, sometimes accompanied by assessment tools of varying complexity [2–4]. More of such models support the existence of at least two dimensions of emotional intelligence: one being inwardly oriented, within one person (having component parts such as self-awareness, self-reliance, the control of one's own emotions, self-actualization, self-sufficiency, conscientiousness, etc.), and the other being outwardly oriented, towards other people (having component parts, such as empathy, communication with others, conflict management, stress tolerance, teamwork, etc.) [2,5,6].

This increasing interest in an in-depth understanding of the concept and its assessment is due to the practical importance of emotional intelligence especially in the area of personal, academic and professional success [1,7–9]. Thus, various research studies have demonstrated the direct correlation between emotional intelligence and job satisfaction, the level of self-actualization, cooperation, physical and mental well-being, employability, the quality of relationships, stress management or self-efficiency, as well as the negative correlation between emotional intelligence and professional stress, anxiety at work, and burnout, respectively [3,4,8,10–13].

Another reason which has contributed significantly to increasing the importance of the concept of emotional intelligence is the generational context greatly marked by the entry into the workforce of a very large number of millennials, the generational cohort with a disruptive profile compared to the previous generations, a phenomenon known as "inter-generational gap" [14,15]. It can be considered that the millennial generation, consisting of individuals born in the period 1982–2004, represents the most researched generational cohort, especially because of the problems generated in the sphere of human resources. There is a very rich specialized literature oriented towards understanding the profile of these individuals, and finding practical solutions for their management [4,15]. The profile of the millennials contains contrasting elements. On the one hand, there are some positive ones, such as social responsibility, civic sense, highly developed teamwork expertise, technical skills, being the first generation of digital natives, resource sharing, respect for diversity, opening to the new, tolerance to diversity [4,15–21]. Contrastingly, the negative aspects include egocentrism, narcissism, superficiality, disloyalty, lack of empathy, lack of involvement and lack of emotional warmth, etc. [4,15–20,22–30]. From the perspective of the present research, the negative elements are very relevant, many of them highlighting the low level of emotional intelligence of the millennial natives. In an attempt to find an explanation for these realities, a possible conclusion has been formulated—the basis of the negative elements in the profile of millennials is a deep lack of self-awareness [28].

In this context, several researchers have carried out studies highlighting the importance of emotional intelligence in education, the intervention in this area being considered vital for the personal and professional success of millennials [1,31,32].

### 1.2. Increasing the Level of Emotional Intelligence through Education

Skeptics in this area have raised the essential question of whether or not emotional intelligence can be taught. In this sense, along with Goleman, other authors support the idea according to which emotional intelligence can be improved and educated, even in adulthood. The training programs focusing on the development of emotional intelligence have proved their efficiency, which, from the perspective of the present research, represents an argument in favor of the introduction in higher education of some curricular contents related to emotional intelligence [3,4,31,33–36].

Because higher education has traditionally concentrated on training and developing hard skills, several authors have recommended the inclusion of soft skills training courses for the development of emotional intelligence within the framework of universities [37]. In this context, more and more specialists have highlighted a phenomenon encountered in education, namely, the transition from hard skills to soft skills [33,38,39]. The specialized literature even mentions the responsibility of higher education teachers to train exactly those soft skills related to emotional intelligence, which millennial students need the most in the workplace, using the most appropriate pedagogical strategies and techniques [13,19]. Recently, it has been argued that the introduction of curricular contents related to emotional intelligence is needed to support millennial students in developing the skills necessary in the context of the COVID-19 pandemic, irrespective of the field in which they study [13]. The above presented aspects are part of a recent tendency supporting the shift from formal instrumental teaching/learning to "transforming" teaching/learning focused on shaping the pupil's/student's personality [13,39–41]. This tendency is supported by research studies

that have demonstrated, for example, the fact that a high level of students' emotional intelligence is a good predictor of professional and personal success, or the strong correlation between the students' academic success and their level of emotional intelligence, or the fact that students with a high level of emotional intelligence tend to complete their studies and be employed in the field of their studies [7,32,42].

The generational context described above raises the issue of rethinking the pedagogical strategies so as to actively interfere in the millennial students' vulnerable areas in order to support them in improving their soft skills and allowing them to use their specific potential.

There are specialists who have highlighted the fact that the specific needs of millennial students are centered on emotions, which makes it necessary to shape and develop their "emotional skills" [33,43]. Some authors have even considered the issue of emotional "literacy", due to their lack of soft skills, millennial students need appropriate training of some basic skills regarding the management of emotions, mainly connected to knowing one's emotions, the awareness of one's own emotions, as well as the others' emotions [34,44]. In this sense, certain pedagogical techniques can be used, including self-assessment, keeping a diary, writing a reflective essay, creative writing, experiential techniques followed by debriefing [31,38,41,44–46]. Recently, researchers have discovered that the use of creative and reflective writing techniques in higher education, irrespective of the field of study, helps students develop soft skills such as empathy, critical thinking, self-awareness, expressing emotions and insight, which can be applied in any field, and play a very important role in the present generational context [47–53]. An important aspect related to the use of expressive writing techniques in higher education refers to their role in building the young students' professional identity, in increasing their professionalism and interest for their future work [40,47,48,53].

The studies presented above speak in favor of incorporating courses focusing on the emotional intelligence development in higher education, and of using appropriate pedagogical techniques to train future responsible employees, who are involved and productive, in the long term, in their field of study and who, at the personal level, succeed in accessing the highest development level in Maslow's pyramid—self-actualization. In the case of the present research, it should be mentioned that some fields have a strong multidimensional and generalized social impact, one of these being the food industry. Training future specialists in the food industry, with a high level of emotional intelligence, responsible both towards themselves and towards others, might be a useful a useful solution to the crisis of the modern world brought about by the depletion of basic resources, consumerism, and the unprecedented proliferation of physical and mental diseases caused by unhealthy nutrition.

### 1.3. The Hypotheses

As indicated in the previous section, the present research emphasizes two components of emotional intelligence, an internal one—the self-awareness of the participating millennial students—and an external one—their level of empathy. The morality level has been the third selected component, as a corollary of the negative aspects present in the millennials' profile. Their lack of morality stems from the fact that they are ego-centered, narcissistic and superficial, and lack empathy and emotional warmth.

The main purpose of the present research has been to highlight the impact of introducing, four modules centered on the development of emotional intelligence within the framework of the Marketing course. These modules are targeted at the millennial generation students, enrolled in two study programs in the field of food engineering. By using some reflective writing tasks, the modules aim to increase the level of students' empathy, awareness of their own emotions and moral identity, thus contributing to the training of a responsible workforce on the labor market in the food industry.

The research hypotheses have been formulated as follows:

**H1.** *If the millennial students enrolled in a study program in the field of food engineering take part in a course which includes elements of emotional intelligence development, based on self-testing and reflective writing techniques, then their empathy level will increase.*

**H2.** *If the millennial students enrolled in a study program in the field of food engineering take part in a course which includes elements of emotional intelligence development, based on self-testing and reflective writing techniques, then the level of awareness of their own emotions will increase.*

**H3.** *If the millennial students enrolled in a study program in the field of food engineering take part in a course which includes elements of emotional intelligence development, based on self-testing and reflective writing techniques, then their moral identity will develop further.*

## 2. Materials and Method

### 2.1. Method

This research falls into the category of psycho-pedagogical experiment because it was carried out in the academic environment and its aim was to study the consequences of introducing modules related to the development of emotional intelligence on the personality of the students enrolled in study programs in the field of food engineering.

To test the hypotheses, the design is quasi-experimental with a single pretest–posttest measurement group. It is a quasi-experimental design because the Marketing course was compulsory for all the 4th year students from Food Engineering (FE) and Food Control and Expertise (FCE), that is, the group is not randomized, and a control group was not used.

The experimental condition presented to the subjects consisted of their participation in the Marketing course, which included four consumer psychology modules with elements of emotional intelligence development, as well as self-testing and reflective writing tasks.

The independent variable (IV), also considered the cause variable, is represented by the four consumer psychology modules. The dependent variables (DV), considered the effect variables, are the empathy level/empathic care (DV.1), the level of awareness of one's own emotions (DV.2), and the level of moral identity (DV.3). The research design is also presented in Table 1 below.

**Table 1.** The Research Design.

|  | Pretest | Experimental Treatment | Posttest |
|---|---|---|---|
| Experimental Group 4th year undergraduate students from FE & FCE | O1 DV.1, DV.2 & DV.3 | X 4 consumer psychology modules | O2 DV.1, DV.2 & DV.3 |

Given that the attendance is not compulsory at the course and seminar activities, there are some students in the experimental group who did not actually attend all the classes and only had at their disposal the materials uploaded on the e-learning platform of Transilvania University of Brașov in order to study the taught aspects, to perform the self-testing and the reflective writing tasks.

Taking into account the fact that 57 valid answers were registered from students who completed the questionnaires both in the pretest and posttest stages, the statistical procedure used was the paired sample *t*-test.

### 2.2. Participants

A total of 69 students participated in the experimental research. The students are enrolled in two study programs in the field of food industry: 39 students in the study program Food Control and Expertise (FCE), out of a total of 40 students, enrolled in the academic year 2021–2022, and 30 students from the study program Food Engineering (FE), out of a total of 31 students, enrolled in the academic year 2021–2022. However, only 57 responses were considered valid, taking into account the research design. Thus,

out of the 39 participating students from the FCE study program, only 29 completed the questionnaire both in the pretest and posttest stages, and out of the 30 students from the FE study program, only 28 completed the questionnaire in both the pretest and posttest stages.

As the research was conducted by the teacher who delivered the Marketing classes, due to the teacher–student force ratio, further measures were taken to make the students feel safe, and to remove any possibility of the teacher identifying the student. The electronic form designed for data collection excluded the transmission of any personal information (name, initials, e-mail address, etc.). Thus, the anonymity of the participants has been ensured. The participation in the research was free/voluntary, and the students had the possibility not to complete the electronic forms transmitted by the teacher. However, as shown in Table 2 below, the participation percentage was very high: in the pretest stage, 58 students out of 71 filled in the electronic form, whereas in the posttest stage, 68 students completed the form. Just one student completed the questionnaire only in the pretest stage and not in the posttest stage. The resulting percentage of voluntary participation was very high, with 97.2% of the students participating in the research.

**Table 2.** Participants.

| | Year of Study | Study Program | Number of Participants | Percentage out of the Total Number of Enrolled Students | Research Stage | Gender | Valid (Pretest and Posttest) |
|---|---|---|---|---|---|---|---|
| | IV | FCE | 39 | 97.5% | Pretest = 30 | M = 5 F = 25 | 29 (72.5%) |
| | | | | | Posttest = 38 | M = 14 F = 24 | |
| | IV | FE | 30 | 96.8% | Pretest = 28 | M = 9 F = 19 | 28 (90.3%) |
| | | | | | Posttest = 30 | M = 11 F = 19 | |
| Group experimental (total) | IV | FCE & FE | 69 | 97.2% | | | 57 (80.3%) |

### 2.3. Materials

The experimental treatment (IV) consisted of four consumer psychology modules designed on three dimensions: teaching, self-testing and reflective writing, which are discussed below.

### 2.3.1. Module 1: Introversion/Extraversion and Nutrition

The module consisted of: 2 hours of teaching the theoretical aspects concerning the introversion/extraversion axis of the human temperament model, including the explanation of the bi-dimensional temperament model, the connection between introversion/extraversion and human nutrition, 2 hours of practical activities (seminar) where the students tested themselves (introversion/extraversion), and a task of reflective writing which was part of their individual study. The reflective writing task was formulated as follows: "On the basis of the taught theoretical aspects and of your test results, write an essay with the topic *I am an introvert/extrovert and that is how I eat*".

### 2.3.2. Module 2: Emotional Stability/Neuroticism and Nutrition

The module included: 2 teaching hours discussing the theory of the axis emotional stability/neuroticism of the human temperament model, the connection between emotional stability/instability and human nutrition, 2 hours of practical activities (seminar) in which the students tested themselves (emotional stability/neuroticism), and performed individually a task of reflective writing, which was formulated as follows: "On the basis of the taught theoretical aspects and of your test results, write an essay with the topic *I am emotionally stable/unstable and that is how I eat*".

### 2.3.3. Module 3: Attachment Patterns and Nutrition

The module consisted of: 2 hours of teaching the theoretical aspects concerning the patterns of secure/insecure attachment, the connection between the type of attachment and human nutrition; 2 hours of practical activities (seminar) in which the students tested themselves (attachment pattern), and a task of reflective writing which was accomplished as an individual study by the students. The reflective writing task was formulated as follows: "Based on the taught theoretical aspects and on your test results, write an essay with the topic *My Attachment Type–My Nutrition*".

### 2.3.4. Module 4: Eating Disorders

The module consisted of: 2 teaching hours concerning the theoretical aspects of eating disorders presented in the DSM-5 Textbook, 2 hours of practical activities (seminar) in which the students tested themselves (eating disorders), and a task of reflective writing which was accomplished as individual study by the students. The reflective writing task was formulated as follows: "On the basis of the taught theoretical aspects and of your test results, write an essay with the topic *If I had an eating disorder, this would be my life*". The writing task asked the students to put themselves empathetically in the place of a person with an eating disorder (of their choice from those taught) and to present its consequences on the life of such a person.

The tests used by the students in self-testing were taken from the ResearchCentral platform (at researchcentral.ro, accessed on 27 September 2022), and all the tools included on this platform can be used freely for research purposes.

The tools used for testing the research hypotheses in the pretest and posttest stages were the following:

1.  Empathic Concern Scale [5]—The scale assesses the level of empathy. It consists of 10 items and is part of the tool called Seven Components Potentially Related to Emotional Intelligence, elaborated by Barchard, and included in the IPIP—International Personality Item Pool Project, initiated by Lewis Goldberg in 1996, project which created an extensive database of over 2000 items for personality assessment.
2.  Attending to Emotions Scale [5]—The scale assesses emotional awareness. It consists of 10 items, and is part of the tool entitled Seven Components Potentially Related to Emotional Intelligence, elaborated by Barchard and included in the IPIP—International Personality Item Pool Project, initiated by Lewis Goldberg in 1996, project which created an extensive database of over 2000 items for personality assessment.
3.  Moral Identity Questionnaire [54,55]—The scale assesses morality as a component part of human personality and consists of 20 items.

All of the three tools are identically scored, with 5-point Likert scales (1—strongly disagree; 5—strongly agree). The three instruments of the present research were taken from the online ResearchCentral portal, which contains tools made freely available to researchers for the purpose of conducting academic research.

### 2.4. Data Collection and Analysis

A questionnaire was created through Google forms. This questionnaire included the three previously mentioned scales and three identification questions (gender, program of study and code name, the same one in both pretest and posttest stages), as well as an informed consent statement. The link was transmitted to the students, in the pretest stage as well as in the posttest one. The fall semester consisted of 14 weeks, the four consumer psychology modules were included starting from the 9th up to the 12th week. Both data collection stages lasted for 5 weeks, the pretest stage took place between 18 November and 23 December, while the posttest stage lasted from 9 January to 13 February.

The students were informed in the pretest stage about the objective pursued by the researchers in order not to be biased in any way. However, they were informed that they were free to complete the form or not, and that the decision to fill in the form or not will not influence in any way the assessment in the Marketing course. In order to ensure

maximum objectivity in the answers and to reduce the examiner's effect, the completion was anonymous, each student using a code name not connected with his/her surname or first name.

Based on the students' self-reported data, the average score of each participant was determined, referring to empathy, the awareness of their own emotions and moral identity, in the pretest stage, as well as in the posttest one. The data were introduced in a database created in SPSS.

Taking into account the quasi-experimental research design for subjects with a single pretest–posttest measurement group, as well as the sample size (57 valid responses), for the testing of statistical hypotheses H1, H2 and H3, a parametric method was used—the *t* test for paired samples. The conditions for applying this test were met: the samples were paired, the dependent variables are quantitative, the valid sample was greater than 30 people and the variables were normally distributed [56].

## 3. Results

The first stage of the research was aimed at the descriptive statistics (Table 3).

**Table 3.** Descriptive Statistics.

| | | *n* | Min. | Max. | M | SD | Skew | Kurt |
|---|---|---|---|---|---|---|---|---|
| Empathic Concern | Pretest | 58 | 2.2 | 4.7 | 3.612 | 0.56 | −0.59 | 0.12 |
| | Posttest | 68 | 2.3 | 5.0 | 3.665 | 0.61 | −0.36 | −0.29 |
| Moral Identity | Pretest | 58 | 2.7 | 5.0 | 4.119 | 0.61 | −0.69 | −0.34 |
| | Posttest | 68 | 3.0 | 5.0 | 4.140 | 0.53 | −0.21 | −0.70 |
| Attending to emotions | Pretest | 58 | 2.5 | 5.0 | 3.895 | 0.65 | −0.06 | −0.69 |
| | Posttest | 68 | 2.8 | 5.0 | 4.081 | 0.60 | −0.56 | −0.54 |

The *t* test was used for testing the three statistical hypotheses, and the obtained results are reported in Table 4 below.

**Table 4.** Descriptive Statistics after using the *t* test.

| | | *n* | M | *t* | *p* | Effect Size r |
|---|---|---|---|---|---|---|
| Empathic Concern | Pretest | 57 | 3.600 | −2.562 | 0.017 | 0.32 |
| | Posttest | 57 | 3.735 | | | |
| Attending to Emotions | Pretest | 57 | 3.875 | −3.560 | 0.001 | 0.43 |
| | Posttest | 57 | 4.098 | | | |
| Moral Identity | Pretest | 57 | 4.104 | −1.276 | 0.207 | 0.17 |
| | Posttest | 57 | 4.193 | | | |

In the case of the students participating in the present research, the level of empathy has increased significantly after taking part in the four consumer psychology modules (mean score 3.735) as compared to the level that existed before (mean score 3.600), t = −2.562, $p < 0.05$, r = 0.32. The obtained results support the acceptance of the alternative hypothesis H1 in the case of empathy. It should also be noted that the effect size (0.32—is in between 0.30 and 0.50) indicates that, beyond the statistical significance, the differences obtained have an average practical importance.

For the students participating in the present research, the level of the awareness of their own emotions has increased significantly after taking part in the four consumer psychology modules (mean score 4.098) as compared to the level that existed before (mean score 3.875), t = −3.560, $p < 0.05$, r = 0.43. The obtained results support the acceptance of the alternative hypothesis H2 in the case of the awareness of their own emotions. It should also be noted that the effect size (0.43—in between 0.30 and 0.50, closer to 0.5) indicates that, beyond the statistical significance, the differences obtained have a practical importance which is average to high.

In the case of the students participating in the present research, the level of moral identity has not increased significantly after taking part in the four consumer psychology modules (mean score 4.193) as compared to the level that existed before (mean score 4.104), t = −1.276, $p$ = 0.207, r = 0.17. The obtained results support the rejection of the alternative hypothesis H3 in the case of moral identity. The effect size is low in this case (0.17 < 0.30); in reality, the obtained results having no practical importance.

## 4. Discussion

As stated in the theoretical framework of the present research, there are multiple previous studies that highlighted the fact that millennials have narcissistic traits, a low level of empathy, and are superficially connected to their own person and to their own emotions [4,15–20,22–30]. The results obtained in the present research in the pretest stage contradict most of the conclusions of these previous studies that support the acute presence of negative traits in the profile of millennials. Thus, in the case of the participating students, the average empathy score in the pretest stage was 3.612 out of the maximum of 5, the score for the awareness of one's own emotions in the pretest phase was 3.895 out of 5, which represents the maximum value, and the moral identity score in the pretest stage was 4.119 out of 5. Thus, the scores registered for empathy and awareness of one's own emotions in the pretest stage can be considered to be average to high, and the moral identity score in the pretest stage can be appreciated as a high one. The pertinent question which arises is: what would be the explanation for this difference? The explanation could be found in the pandemic context of the recent years. Thus, it is appreciated that the recent COVID-19 pandemic determined a reevaluation of the role of emotional intelligence in the work field, with the importance of components, such as empathy or awareness, and emotional regulation, respectively, becoming very high [57]. Recent studies have demonstrated the fact that the COVID-19 pandemic led to increases in scores on tests which measure the emotional intelligence level in the case of the teaching staff within the framework of higher education [58]. In the context of the high scores obtained by the participants in the present study, atypical scores for the millennial generation, we are talking about understanding to what extent this change has been determined by the pandemic context. In addition to the above, it can also be highlighted that the total percentage of students participating in the research has been very high, over 95%, despite the total freedom offered to participate or not, and the anonymity ensured by the procedure of data collection. This aspect supports the high average score obtained by the students participating in the moral identity questionnaire. Considering the need for a substantial increase in the responsibility of young millennials, this idea deserves a more in-depth investigation in the future.

As for the interpretation of the results obtained for testing the three formulated statistical hypotheses, the alternative hypotheses H1 and H2 are accepted, going through the four psychology modules has increased the level of empathy (H1) and awareness of one's own emotions in the case of the participating students (H2). In addition to the fact that the results are statistically significant at a significance threshold of less than 0.05, the practical importance of the results should also be noted. Thus, the power of the research is high in the case of the awareness of one's own emotions and medium in the case of empathy, the results obtained are considered valuable. It can be concluded that the prediction formulated in the first hypothesis is supported by data: *If the millennial students enrolled in a study program in the field of food engineering take part in a course which includes elements of emotional intelligence development, based on self-testing and reflective writing techniques, then their empathy level will increase* (H1 confirmed). Additionally, data support the prediction formulated in the second hypothesis: *If the millennial students enrolled in a study program of study in the field of food engineering take part in a course which includes elements of emotional intelligence development, based on self-testing and reflective writing techniques, then the level of awareness of their own emotions will increase* (H2 confirmed). The two constructs, the awareness of one's own emotions and empathy, respectively, lie at the basis of the complex

construct of emotional intelligence, which leads to the conclusion that the objective of the present research has been achieved: the four psychology modules completed by the students enrolled in the two study programs in field of food industry contributed to the increase in the students' emotional intelligence level. It should be emphasized that other research studies with similar objectives, in which university courses were designed to develop students' emotional intelligence, obtained similar results in the sense that the scores on emotional awareness and empathy have increased, more or less significantly, after completing the course, especially for the students with low initial scores [4,59].

In this context, we are talking about identifying those elements which have brought a major contribution to the increase in the empathy and emotional awareness scores of the participating students. A first such strong point of the four psychology modules design can be connected to the students' self-testing activities. Supporting this idea, we can mention similar research studies carried out previously, highlighting the importance of certain self-assessment techniques in academic courses focused on the development of emotional intelligence among millennial students [41,59]. Another idea might be related to the use of writing techniques. Previous research argued that, irrespective of the field of study, they support students in developing soft skills, such as empathy, critical thinking, self-awareness, expressing emotions and insight [47–53]. It should also be mentioned that the specialized literature emphasized the fact that writing techniques must be taught by teachers trained in creative writing [49]. This condition was met, because the teacher has international training in psychotherapy through creative writing. In the same direction, we can mention a series of studies which have emphasized the fact that teaching millennial students can be a rather difficult task because they are very demanding and very superficial at the same time [30,43]. Despite their superficiality, millennial students want teachers who are passionate about what they teach, challenging them during the classes, turning learning into an interactive process, which challenges teachers to authentically interact with the students, emotionally connect with them, and find creative ways to convey the information content within the framework of the taught subjects [30,43,60,61]. In this sense, in the current research, the double specialization of the teacher, in marketing and clinical psychology, as well as the training in psychotherapy, have represented factors with a high contribution in obtaining good results after a relatively short intervention of only four modules. Thus, it is recommended for those teaching staff members involved in teaching curricular contents aimed at developing the students' emotional intelligence, irrespective of the study program, to also have previous training in psychology/psychotherapy.

As observed in the specialized literature, there are components of emotional intelligence that are easier to educate, the example provided being emotional awareness [3]. The results obtained in the present research support this conclusion. It has also been appreciated that empathy is easily teachable [62,63], and the results of the present research support this Although the experiment described in the present research contains only four modules, not an entire course, as it was the case in other previous research studies, the results are consistent even after four course modules. A possible conclusion concerns the importance of the triarchic structure that the teacher created for the four modules: teaching–self-testing–reflective writing. This very structure, as well as the four selected topics, can constitute the original contribution of this experiment, allowing replication in future research.

Another interesting aspect can be noticed in Table 2 (Section 2.2 above), namely that in the posttest stage there was an increase in the number of responses, especially among the male students, a high percentage of them choosing not to participate in the pretest phase of the research. Additionally, experimental mortality has been very low, only one female respondent chose not to participate in the research in the posttest stage. These aspects support the idea that completing the activities in the four psychology modules introduced in the Marketing course had positive effects on the participating students. Thus, the students became more open and responsibly involved in the interaction with the teacher and in the process of self-knowledge.

### 5. Conclusions and Research Limitations

The effectiveness of the introduction of consumer psychology modules in the academic year 2021–2022, the fall semester, in terms of increasing the level of emotional intelligence, and mitigating the negative traits specific to the millennial profile, in the case of students enrolled in the two study programs in food industry (food engineering) at the Faculty of Food and Tourism of Transilvania University of Brasov, has been evaluated on the basis of quantitative data, gradually assessing the following aspects:

(1) The extent to which the consumer psychology modules (which included teaching theoretical aspects, psychological self-testing of students and performing reflective writing tasks) contributed to the development of emotional self-awareness and empathy among the participating students. In this sense, the scores obtained by students on the scales "Attending to emotions" and "Empathic concern" have been interpreted before and after the actual performance of the activities.

(2) The extent to which the consumer psychology modules (which included activities for teaching theoretical aspects, psychological self-testing of students and performing reflective writing tasks) contributed to the reduction in the negative traits specific to the psychological profile of millennials to the extent of determining a reduced level of morality of these individuals. In this sense, the scores obtained by the students on the scale "Moral identity" have been interpreted before and after the actual performance of the activities.

Based on the results obtained, we can draw the conclusion that the main objective of the present experiment has been largely achieved. After completing the activities included in the four consumer psychology modules, strongly focused on the development of the students' emotional intelligence, both the level of empathy and the awareness of their own emotions increased in the case of the participating students. This increase was statistically significant, the results having an average practical importance in the case of empathy, and a high one in the case of the awareness of one's own emotions. Referring to the first two hypotheses formulated for the present research, it can be concluded that both H1 and H2 are backed by the data. Therefore, the results support the formulation of the following hypothesis: *If the millennial students enrolled in a study program of study in the field of food engineering take part in a course which includes elements of emotional intelligence development, based on self-testing and reflective writing techniques, then their empathy level will increase* (H1 alternative hypothesis confirmed). Additionally, results allow us to state that: *If the millennial students enrolled in a study program of study in the field of food engineering take part in a course which includes elements of emotional intelligence development, based on self-testing and reflective writing techniques, then the level of awareness of their own emotions will increase* (H2 alternative hypothesis confirmed).

In the case of the morality level, the scores also increased after the experimental treatment, but this increase was not statistically significant. Therefore, regarding the third research hypothesis formulated in this study, the null hypothesis will be accepted, stating that *there is no effect on level of moral identity of the student's participating in a course which includes elements of emotional intelligence development, based on self-testing and the reflective writing techniques* (H3 alternative hypothesis rejected). For the morality level, the practical importance of the obtained results, or the power of the research, is low, a fact that supports further investigation of the morality level of millennial students, as well as of pedagogical tools that can be used to increase it. In the present study, the researchers started from the assumption that this complex construct—the level of morality—is closely related to the negative aspects highlighted in the profile of millennials—self-centeredness, narcissism, lack of warmth and empathy, etc. The results obtained both in the pretest and in the posttest stage support the idea that this aspect represents a limitation of the present research, thus, requiring more in-depth analysis of the morality construct and its connection with the negative elements in the millennials' profile.

In order to highlight the practical importance of the self-testing activities of and of the reflective writing tasks in increasing the level of empathy and emotional awareness, it is

highly recommended to use a different research design, with a non-randomized control group, in both the pretest and posttest stages, in which only the experimental group will go through the writing tasks. Additionally, a mixed, qualitative and quantitative design can be considered, in which the qualitative data are also analyzed based on the essays written by students. The qualitative data can prove most valuable in highlighting the usefulness of introducing reflective/creative writing techniques in higher education, in study programs that are not related to the existing literature.

Summarizing the main limitations of the present research, we can refer to:

- The research design—without control group, only quantitative;
- The lack of clarity/depth regarding the concept of morality;
- The researchers' subjectivity—Even if the research design is quantitative, one of the researchers was the teacher who actually taught the four consumer psychology modules to the students in the experimental group. As members of the X generational cohort, both researchers greatly value emotional intelligence, empathy, deep self-understanding and morality. Therefore, it can be considered that the entire research approach might have been influenced by the researchers' values and perspectives. This influence can be seen within the formulation of the three research hypotheses. In order to reduce the researcher subjectivity, the four consumer psychology modules have been taught from a neutral perspective, with a focus on the process of (self)discovery undergone by each student.

Several suggestions for future research directions can be formulated:

- Based on the high transferability of the present research design, further research studies can be carried out in different fields of study, mainly in those where soft skills are valuable assets for future employees;
- Because the present design is not very strong, the research study can be replicated with a better research design, which might refer to the use of a control group and/or the combination of quantitative and qualitative data;
- Due to the fact that this research work focused only on two components of emotional intelligence (empathy and emotional self-awareness), taking into account that this concept is a much more complex one, future research studies can go deeper, and introduce into the analysis other facets of emotional intelligence.

In conclusion, the main practical implications of the present research refer to the support of the introduction of curricular contents of psychology, focused on the development of emotional intelligence, within the study programs in the food industry, which should contain both theoretical components and, especially, practical components, which support the students' self-knowledge and reflective efforts. Despite the previously presented limitations, an important practical implication is the use of reflective writing techniques, which has proven to be useful in the study programs that are not related to the linguistic field, particularly contributing to the increase in the students' emotional self-consciousness and empathy level.

**Author Contributions:** Conceptualization: O.-A.G.-P. and L.C.; methodology: O.-A.G.-P. and L.C.; data curation: L.C.; project administration: L.C.; data processing: L.C.; writing—original draft preparation: O.-A.G.-P. and L.C. All authors have read and agreed to the published version of the manuscript.

**Funding:** This research received funding for publication of the research results from Transilvania University from Brasov—Decision number 4984/21.10.2022.

**Institutional Review Board Statement:** The study was conducted in accordance with the Declaration of Helsinki. Approval was obtained from the Department of the Faculty of Food and Tourism, Transilvania University from Brasov—Decision Rep. no.5 from 13 September 2021.

**Informed Consent Statement:** Participants were informed that they were free to choose NOT to participate in the research. Informed consent was obtained from each student who chose to complete the self-testing questionnaire. The present research is part of the category of pedagogical experiments, the purpose of self-testing not being revealed to the students in the pretest stage, in order not to

influence their behavior during the experimental treatment stage and not to influence their results in the posttest stage. All students had access to the main aim, the hypotheses, as well as the results of the present research after the posttest stage was completed.

**Data Availability Statement:** The research database used in the present study is available on request from the corresponding author. The database is not publicly available.

**Acknowledgments:** The authors would like to thank the participating students for their active and responsible involvement during all the stages of the present research project.

**Conflicts of Interest:** The authors declare no conflict of interest.

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
