# Peer review of "Developing the Emotional Intelligence of Millennial Students: A Teaching Strategy"

_sustainability, doi:10.3390/su142113890_

Round 1

Reviewer 1 Report

Dear authors,

Congratulations for your article! The topic is very interesting. I have just some suggestions:

- Please pay attention to repetition. Fo example, On the one hand, there are 110 some positive ones, such as: social responsibility, civic sense, highly developed teamwork 111 skills, technical skills, being the first generation of digital natives, resource sharing, respect 112 for diversity, opening to the new, tolerance to diversity [4, 15, 16, 17, 18, 19, 20, 21]. On the 113 other hand, there have been identified negative aspects, such as: egocentrism, narcissism, 114 superficiality, disloyalty, lack of empathy, lack of involvement and lack of emotional 115 warmth etc. [4, 15, 16, 17, 18, 19, 20, 22, 23, 24, 25, 26, 27, 28, 29, 30].

- Point out better the limitations at the end.

- Emphasise in the conclusion and discussion the connection to each hypothesis 

- Reiterate the subjectivity of the researcher 

Author Response

Dear Madam, Dear Sir,

Thank you for your Review Report, for all your positive and encouraging considerations, as well as for all your pertinent and useful suggestions!

We will do our best to explain in the following paragraph the way we approached each suggestion you made in your Review Report:

  1. We solved the problem related to the repetitions you have indicated.
  2. We wrote a distinct paragraph related to limitations in the final part of the paper. We have included the researcher subjectivity within the limitations of our research endeavor.
  3. We emphasized better the three hypothesis and we linked them to the obtained results/data, in the discussion and conclusions sections of the paper.

We did our best to approach all your suggestions and we sincerely trust you appreciate the new developments included in our paper! We feel greatly encouraged due to the fact that you found the topic as being very interesting!  

Sincerely,

Assoc. Prof. Laura Cismaru

Lect. Oana-Andreea Ghita-Pirnuta

Reviewer 2 Report

The study starts from a social need and the theme is applicable to the context studied and to any other context, I value the transferability of the work. It has been a success to work empathy and emotional awareness from this perspective.

As an area of ​​majority, I would propose expanding a control group to make comparisons with the experimental group, although understanding that the study has been carried out and does not consider it insufficient, I simply encourage this research to be expanded in future works.

I consider the application of pedagogical techniques such as self-assessment and reflective writing to be of great interest, since it recovers the benefits of graphology and invites calm assessment.

I propose to add a small paragraph with the proposals of lines in which it could be investigated to deepen and complement these results

congratulations on the job

Author Response

Dear Madam, Dear Sir,

Thank you for your Review Report, for all your positive and encouraging considerations, as well as for your useful suggestions!

We will do our best to explain in the following paragraph the way we approached each suggestion you made in your Review Report:

  1. We emphasized the fact that a future research design should include a control group.
  2. We wrote a distinct paragraph related to the suggestions which can be formulated for future research directions, in order to deepen and complement the results we have obtained.

We did our best to approach your suggestions and we sincerely trust you appreciate the new developments included in our paper! We feel greatly encouraged due to the fact that you found our work as being highly transferable to other domains!  

Sincerely,

Assoc. Prof. Laura Cismaru

Lect. Oana-Andreea Ghita-Pirnuta

Reviewer 3 Report

I find the topic interesting, and the article is well structured, with a logical sequence of ideas.

Recommendation:

A brief presentation in the Introduction section of the current situation referred to by the authors, namely the unprecedented proliferation of physical and mental diseases caused by unhealthy nutrition and eating habits, would be appropriate to argue the importance of the research carried out.

Although the style of expression is clear and coherent, the article must be checked and corrected. Some phrases require reformulation; they have disturbing repetitions or are difficult to follow (p. 2, lines 53, 54, 55 all contain "in the field", p. 3, lines 141 and 143 "pandemic context" and "COVID- 19 pandemic").

Minor correction for in-text references, when several works are listed successively (p. 2, line 86 instead of [2, 3, 4] is [2-4], p. 3, line 116, instead of [ 4, 15, 16, 17, 18, 19, 20, 22, 23, 24, 25, 26, 27, 28, 29, 30] is [4, 15-20, 22-30], etc.).

Author Response

Dear Madam, Dear Sir,

Thank you for your Review Report, for all your positive and encouraging considerations, as well as for your useful suggestions!

We will do our best to explain in the following paragraph the way we approached each suggestion you made in your Review Report:

  1. We wrote a distinct paragraph in the beginning of the paper in order to better explain to the reader what we referred to when we wrote about “the unprecedented proliferation of physical and mental diseases caused by unhealthy nutrition and eating habits”.
  2. We solved the problem related to the repetitions you have indicated to us.
  3. We did the corrections for the in-text references, when more works were listed successively.

We did our best to approach all your suggestions and we sincerely trust you appreciate the new developments included in our paper! We feel greatly encouraged due to the fact that you found the topic as being interesting!  

Sincerely,

Assoc. Prof. Laura Cismaru

Lect. Oana-Andreea Ghita-Pirnuta
